# Individualized Short-Term Electric Load Forecasting Using Data-Driven Meta-Heuristic Method Based on LSTM Network

**DOI:** 10.3390/s22207900

**Published:** 2022-10-17

**Authors:** Lichao Sun, Hang Qin, Krzysztof Przystupa, Michal Majka, Orest Kochan

**Affiliations:** 1Computer School, Yangtze University, Jingzhou 434023, China; 2Department of Automation, Lublin University of Technology, Nadbystrzycka 36, 20-618 Lublin, Poland; 3Department of Electrical Engineering and Electrotechnology, Lublin University of Technology, Nadbystrzycka 36, 20-618 Lublin, Poland; 4School of Computer Science, Hubei University of Technology, Wuhan 430068, China; 5Department of Measuring Information Technologies, Institute of Computer Technologies, Automation and Metrology, Lviv Polytechnic National University, 79013 Lviv, Ukraine

**Keywords:** short-term load forecasting, meta-heuristic optimization technology, logistic chaos operator, multi-level regulation factor, sine cosine algorithm, recurrent neural network

## Abstract

Short-term load forecasting is viewed as one promising technology for demand prediction under the most critical inputs for the promising arrangement of power plant units. Thus, it is imperative to present new incentive methods to motivate such power system operations for electricity management. This paper proposes an approach for short-term electric load forecasting using long short-term memory networks and an improved sine cosine algorithm called MetaREC. First, using long short-term memory networks for a special kind of recurrent neural network, the dispatching commands have the characteristics of storing and transmitting both long-term and short-term memories. Next, four important parameters are determined using the sine cosine algorithm base on a logistic chaos operator and multilevel modulation factor to overcome the inaccuracy of long short-term memory networks prediction, in terms of the manual selection of parameter values. Moreover, the performance of the MetaREC method outperforms others with regard to convergence accuracy and convergence speed on a variety of test functions. Finally, our analysis is extended to the scenario of the MetaREC_long short-term memory with back propagation neural network, long short-term memory networks with default parameters, long short-term memory networks with the conventional sine-cosine algorithm, and long short-term memory networks with whale optimization for power load forecasting on a real electric load dataset. Simulation results demonstrate that the multiple forecasts with MetaREC_long short-term memory can effectively incentivize the high accuracy and stability for short-term power load forecasting.

## 1. Introduction

Smart grid (SG) is a new type of power system that has emerged in recent years and is widely used by power companies due to its accuracy in power load forecasting [1,2,3,4]. Energy issues are very important nowadays especially with the spread of appliances and the concepts of the Internet of Things [5,6], smart house [7], and smart city [8]. Thus, the problem of energy efficiency is one of the key ones. That is why a lot of attention is paid to reliable energy sources for domestic use [9,10,11] to maintain the required infrastructure. There are also many energy saving techniques [12,13]. However, the main problems of energy efficiency belong to energy generation and transfer. In particular, inaccurate sensors in producing facilities [14,15]. These inaccuracies have considerable effect on the world economy. According to the estimation [16], yearly losses are about USD 400 million worldwide, therefore many studies are carried out in this field [17,18]. In particular, new sensors [15,17] and techniques for data processing were proposed [17,19]. However, proper operation of SG is required, which is why a lot of attention has focused on this field. One of the key problems in the field of proper operation of SG is the short-term forecast of load. Short-term electric load forecasting usually refers to making forecasts of power loads for the next few hours or days. It guarantees the safe, energy-saving, and efficient operation of the power system, and has a vital role in daily life [20,21,22]. For the purpose of enhancing the efficiency of regional power grids in distribution dispatching and to meet the power demand from the grid center to each region, there is a need to acquire higher forecast accuracy for short-term power load forecasting. There are many factors that can affect electric load forecasting, including meteorological factors, seasonal changes, and time phases [23,24]. In addition, the stochastic and non-linear nature of electricity consumption by households is also an important consideration that affects electricity load forecasting. Fortunately, in recent years, with the advent of more sophisticated measurement instruments and recording devices, researchers can more easily document and analyze the effects of these external factors on changes in electric load data. As a result, two main types of electric load forecasting methods have emerged: the traditional forecasting methods and machine learning-based forecasting ones [25,26]. The traditional forecasting methods are characterized by use of time-series relationships in historical load data to calculate forecasts [27]; and common traditional forecasting methods are exponential smoothing [28], Kalman filtering [29], and auto-regression [30] methods. The limitation of traditional methods is that they have limited prediction capability for non-linear relationship data and are not suitable for predicting short-term electric loads with high randomness and complexity. Machine learning-based prediction methods include support vector machines, random forests, convolutional neural networks, and so on [31,32,33]. Most of these methods are used for short-term power load forecasting that requires manual setting of temporal features and needs to fully consider the characteristics of data in terms of temporality. Among them, the long and short-term memory neural network (LSTM) is a modified version of a recurrent neural network (RNN) in machine learning, which can make full use of the temporal characteristics of data and make efficient predictions for data with strong nonlinear and temporal characteristics [34,35]. However, the LSTM is comparable to other neural networks in light of parameter selection, and the selection of its model parameters mainly relies on manual selection based on previous experience [36]. In addition, the values of the parameters can have a significant impact on the performance of the model.

In the last two decades, electric load forecasting models based on neural networks and population intelligence algorithms have been proposed to solve the above-mentioned problem. The population intelligence algorithm is a common meta-heuristic optimization [37]. Compared with traditional methods, the population intelligence algorithm avoids calculating gradients. It is essentially a probability-based search algorithm, which has the advantages of fast search speed and efficient search capability for complex optimization problems [38]. Common population intelligence optimization algorithms include particle swarm optimization (PSO) [39], the firefly algorithm (FA) [40], the whale optimization algorithm (WOA) [41], the sine cosine algorithm (SCA) [42], etc. SCA generalizes and absorbs the iterative strategies of some of the swarm intelligence optimization algorithms, grouping the optimization-seeking iterative strategies of many intelligence optimization algorithms into two parts: global search and local exploitation, with the advantages of fewer parameters and easier implementation.

In this regard, in order to obtain highly accurate short-term electricity load forecasts, a short-term electric load forecasting method based on the improved sine cosine algorithm and LSTM model, called MetaREC is proposed. The main contributions of this paper are shown below.
A hybrid forecasting framework based on the improved SCA and LSTM is proposed and it is used to solve the problem of short-term electric load forecasting;Logistic chaotic operators and multilevel modulation factors are used to overcome the problem of the conventional SCA tending to fall into local optima during the optimization process;The parameter fetching problem of the LSTM by the improved SCA is optimized and then the optimized LSTM is used to forecast the short-term electric load;The method proposed in this paper for short-term electricity load forecasting is used, and the experiments demonstrate high efficiency of the method.

The rest of this paper is organized as follows. Section 2 briefly describes the related works. The description and definition of the problem are given in Section 3. Further, the model for forecasting the electric load and the process of using the model for electric load forecasting is given in Section 4. In Section 5, the proposed forecasting model is used to forecast real power load data and evaluate the performance of the forecasting model. Finally, the conclusions are given in Section 6.

## 2. Materials and Methods

Short-term electric load forecasting is critical to our lives. Electricity load forecasting models are mainly divided into traditional forecasting models and forecasting models based on machine learning methods. Researchers have conducted a lot of research to improve the accuracy of forecasting. In previous studies, in order to more clearly describe the previous studies on electric load forecasting, these studies were divided into single forecasting models and hybrid integrated forecasting ones. The specific discussion is shown below.

A single model is defined as a forecasting model that is used only to forecast the electric load. In [43], the authors proposed a multilayer bidirectional recurrent neural network based on the LSTM and the gated recurrent unit (GRU). They used this network for short-term power load forecasting, and compared this method with the LSTM, SVM, and BP on two data sets. The comparison of the prediction results showed the superiority of the method proposed in [43]. An improved exponential smoothing gray model based on this model is proposed in [44], and this model is applied to short-term power load forecasting. The method not only improves the accuracy of short-term power load forecasting but also extends the application scope of gray forecasting. In [45], a radial basis function neural network (RBFNN) is used in the prediction of electric load, and the results show that it has considerable accuracy and stability in the prediction of electric load. An autoregressive integrated moving average (ARIMA) and support vector machine (SVM)- based power forecasting method was proposed [46]. Its core idea is to first use ARIMA to forecast the daily load and then use SVM to correct the previously obtained forecast deviations. The experimental results show the high accuracy of the method for forecasting large sample electricity data. A model using dynamic neural networks for power load forecasting was presented in [47] and the structure of the neural network was also validated in the paper using regression plots. The experimental results show the efficiency of the method in [47] for forecasting complex time series. The authors in [48] proposed a support vector regression (SGA-SVR) based on a sequential grid approach for forecasting electric loads. In the experimental results, the SGA-SVR showed considerable prediction performance. 

A hybrid integrated prediction framework uses feature extraction or optimization models mixed with prediction models to achieve improved prediction accuracy. A hybrid power forecasting framework based on SVM and ant colony optimization was proposed in [49]. The authors compared the performance of this method with SVM and BP on short-term power loads and the results showed that this method can achieve better prediction accuracy. In [50], a hybrid power forecasting method based on the generalized regression neural network (GRNN) and the fruit fly optimization algorithm (FOA) was proposed. The authors used the FOA to solve the problem of how to select the appropriate propagation parameters in the GRNN. Finally, this method is compared with a variety of other forecasting methods for prediction performance, and the experimental results prove the effectiveness of the method [50]. A hybrid power forecasting method based on the least squares support vector machine (LSSVM) and the moth-flame optimization algorithm (MFO) was proposed [51]. The authors in [36] used MFO to determine two parameters (σ and C) in the LSSVM model. The prediction method combining second-order oscillations and repulsive particle swarms used to optimize SVM parameters was proposed in [52] and applied to power data prediction in Singapore. In [53], the power load forecasting method combining the differential evolution (DE) algorithm with SVR is proposed and the forecasting performance of this method is compared with SVR with default parameters, BP, artificial neural network (BPNN), and regression forecasting methods. The experimental results show the efficiency of the method in [53]. Consequently, [54] used global optimal particle swarm optimization (GPAO) to improve the prediction accuracy of feedforward artificial neural networks (ANNs) and tested this model on ISO New England grid data. The test results demonstrated the prediction accuracy of the method. A wavelet neural network (WNN) hybrid electric load forecasting model based on improved empirical modal decomposition (IEMD), autoregressive integral shift (ARIMA), and fruit fly optimization algorithm (FOA) was proposed in [55]; the simulation results then show that the model has good performance in power load forecasting.

Although all the above forecasting methods in the literature have achieved different degrees of improvement in electric load forecasting, it was found that few researchers have applied the LSTM forecasting model with optimized parameters by the sine cosine optimization algorithm (SCA) to electric load forecasting in previous studies. The LSTM can perform well in long-sequence problems, moreover, the SCA optimization is also an algorithm with considerable optimization effect. The purpose of this paper is to propose a power load forecasting model based on the improved SCA with optimized LSTM parameters, in order to achieve higher forecasting accuracy.

## 3. Problem Formulation

In this section, the short-term electric load forecasting problem is defined from the mathematical point of view and delivers an optimized forecasting framework.

The process of electric load forecasting is to first split the historical power load data into training and testing sets, then to train the fitted prediction model with the training set, and finally to use the fitted prediction model to validate the prediction on the testing set. The main symbols and their meanings in this paper are listed in Table 1. 

Assuming that n is the number of samples in the testing set, the optimization problem of electric load forecasting can be defined as follows:(1)$$Minimize:\;f\left(x\right)=\frac{1}{n}\overset{n}{\underset{i=1}{\sum }}{\left(y_{i}-\bar{y_{i}}\right)}^{2}.$$
$$subject\;to:y_{i}\geq 0\;,i=1,\;2,\;3,\;\cdots ,n,\;y_{i}\in Y,i=1,\;2,\;3,\;\cdots ,n\;\;L_{j}\leq x_{j}\leq U_{j},\;j\in \left[1,4\right]$$
where x is a solution in the optimization algorithm, it is a matrix with four columns in a row, and the four elements of the matrix represent the learning rate, the number of training sessions, the number of neurons in the first layer, and the number of neurons in the second layer in the LSTM network, respectively. Let $L_{j}$ and $U_{j}$ represent the upper and lower bounds of the j−th parameter, respectively. Let $f\left(x\right)$ be the fitness value of the solution x and the mean square error of the optimized LSTM on the testing set. Let n be the number of samples in the testing set, and Y represents the testing set. Let $y_{i}$ denote the real power load data of the i−th time period. Let $\bar{y_{i}}$ be the forecast power load data for the i−th time period. The short-term power load forecasting framework in this paper is shown in Figure 1, and the MetaREC method will be described in detail in Section 4.

## 4. Stochastic Power Forecasting with Data-Driven Heuristic Method

### 4.1. Meta-Heuristic Method

#### 4.1.1. Standard Sine Cosine Algorithm

The essence of the sine cosine algorithm (SCA) is to find the optimal value by using the perturbation properties of the sine and cosine functions [56,57,58]. Contrasted with other meta-heuristic algorithms, the advantages of the SCA are as follows: fewer parameters and a simpler structure. The optimization search process of the SCA can be split into three steps as follows.

Step 1. Determination of initial populations

The initial population is calculated according to the following:(2)$$\begin{array}{c} \;X\left(i,j\right)=X\left(min,j\right)+random\left(0,1\right)\times \left(X\left(max,j\right)-X\left(min,j\right)\right) \end{array}$$
where $X\left(max,j\right)$ and $X\left(min,j\right)$ are the upper and lower limits of the individual on dimension j, respectively. Let R be a random number within $\left(0,\;\;1\right)$. 

Step 2. Calculation of amplitude factor, $r_{1}\left(t\right)$, and random numbers $r_{2},\;r_{3},\;\mathrm{and}\;r_{4}$.

The amplitude factor is the key part to control the SCA to convert between global search and local search, and the update formula of the amplitude factor is shown below
(3)$$\begin{array}{c} \;\;\;\;\;\;\;r_{1}\left(t\right)=\alpha \times \left(1-\frac{t}{T}\right),\;\;\;\alpha =2\;. \end{array}$$

The parameters $r_{2},\;r_{3},\;\mathrm{and}\;r_{4}$ are random numbers, each obeying uniform distribution,
(4)$$\begin{array}{c} \left\{\begin{array}{c} r_{2}\in \left[0,\;\;\;2\pi \right]\\ r_{3}\in \left[-2,\;\;\;2\right]\\ r_{4}\in \left[0,\;\;\;\;\;\;1\right] \end{array}\right.. \end{array}$$

α is specified as a constant and generally is Equation $\left(3\right)$. Let T be the total number of iterations of the algorithm in the optimization search process.

Step 3. Renewal of populations.

The population is updated according to the following formula.
(5)$$\begin{array}{c} \;\;\;\;\;\;\;\;\;\;\;\;\;\;\;\;\;\;\;\;\;\;\;\;\;\;X_{t+1}\left(i,j\right)=\left\{\begin{array}{c} X_{t}\left(i,j\right)+r_{1}\left(t\right)\times sin\left(r_{2}\right)\times \left|r_{3}\times P_{t}\left(i,j\right)-X_{t}\left(i,j\right)\right|,\;\;\;r_{4}<0.5\\ X_{t}\left(i,j\right)+r_{1}\left(t\right)\times cos\left(r_{2}\right)\times \left|r_{3}\times P_{t}\left(i,j\right)-X_{t}\left(i,j\right)\right|,\;\;\;r_{4}\geq 0.5 \end{array}\right.. \end{array}$$
where $X_{t}\left(i,j\right)$ denotes the position of individual i at the time when the number of iteration rounds is t; and $P_{t}\left(i,j\right)$ represents the optimum position achieved by the particle in the previous t iterations.

The SCA has been successfully used in many fields based on its efficient merit-seeking capability. However, thus far, there has been little research conducted using optimized SCA to solve optimum production schemes for electricity. Therefore, in this paper, the improved version of the SCA is proposed for short-term power load forecasting.

#### 4.1.2. Modified Sine Cosine Method

The conventional SCA uses a random initialization strategy in the population initialization phase, which leads to problems such as the algorithm can easily fall into local optima and slow convergence speed. However, the introduction of the chaos operator can improve this drawback. Chaos is a phenomenon in which deterministic systems spontaneously produce instability. Under the three characteristics, i.e., randomness, regularity, and ergodicity, the chaotic motion can traverse all states without repetition in a certain range and according to its own laws. Therefore, using chaotic variables to handle optimization search is far more efficient than using blind and disorderly random searches [59,60,61]. 

There are various types of chaotic variables, and in this paper, the logistic chaos operator is used, whose expression is shown below. The logistic chaos mapping is shown in Section 4.2.1.
(6)$$\begin{array}{c} X\left(t+1\right)=\mu \cdot X\left(t\right)\cdot \left(1-X\left(t\right)\right),\;\;0\leq \mu \leq 4 \end{array}$$
where t is the number of iteration steps, and for any t there is $X\left(t\right)\in \left[0,\;1\right]$. 

When μ is taken in different ranges, the results of the logistic system appear in three different states: stable point, periodic, and chaotic. When the parameter μ is in the range of $\left[3.57,\;4\right]$, the motion trajectory of logistic shows chaotic characteristics. When μ=3.57, the logistic appears chaotic. The main characteristic of chaos is that even small changes in the initialized populations lead to significant differences in the results as time increases. As μ continues to increase, the results of the logistic iteration switch between periodic and chaotic types. When u=4, the logistic results will be completely chaotic, which will eventually lead to a uniform distribution of the results on the interval $\left[0,\;1\right]$. In this paper, logistic operator with u=4 is chosen for the experiments.

In addition, the value of the adjustment factor, $r_{1}\left(t\right)$, decreases linearly with the number of iterations in the conventional SCA. According to the literature [62], it is known that the SCA will converge faster when $r_{1}\left(t\right)<1$. If the SCA falls into a local optimum at this point, the result of the SCA will become unsatisfactory. To solve this problem, this paper uses a multilevel adjustment factor, $r_{1}^{*}\left(t\right)$, to change the value of the adjustment factor according to the change in the number of iterations. The multilevel adjustment factor, $r_{1}^{*}\left(t\right)$, was set to four different equations, as shown in Figure 2.

The multilevel regulatory factor is defined in Equation (7). The number of iteration rounds was split into four phases with $T_{1}=\left[0,\;\left.\frac{1}{4}T\right),\;T_{2}=\left[\frac{1}{4}T,\;\left.\frac{1}{2}T\right),\;T_{3}=\left[\frac{1}{2}T,\;\left.\frac{3}{4}T\right),\;T_{4}=\left[\frac{3}{4}T,\;\left.T\right]\right.\right.\;\right.\right.$.
(7)$$r_{1}^{*}\left(t\right)=\{\begin{array}{cc} a\times {\left(\frac{1}{2}\right)}^{1-\frac{t}{T_{1}}} & ,t\in \left[0,\frac{1}{4}T\right)\\ \tan h\left[a\times \left(1-\frac{t-T_{1}}{T_{2}-T_{1}}\right)\right] & ,t\in \left[\frac{1}{4}T,\frac{1}{2}T\right)\\ a\times {\left(\frac{1}{2}\right)}^{1-\frac{t-T_{2}}{T_{3}-T_{2}}} & ,t\in \left[\frac{1}{2}T,\frac{3}{4}T\right)\\ \tan h\left[a\times \left(1-\frac{t-T_{3}}{T_{4}-T_{3}}\right)\right], & ,t\in \left[\frac{3}{4}T,T\right) \end{array}$$

Substituting a=2 into $a\times {\left(\frac{1}{2}\right)}^{1-\frac{t}{T}}$ and $tanh\left(a\times \left(1-\frac{t}{T}\right)\right)$ gives the following result.
(8)$$\begin{array}{c} \;\;\;\;\;\;\;\;\;\;\;\begin{array}{c} \underset{t\to 0}{\lim}\left[2\times {\left(\frac{1}{2}\right)}^{1-\frac{t}{T}}\right]=\underset{t\to 0}{\lim}{\left(\frac{1}{2}\right)}^{\frac{t}{T}}=1,\\ \underset{t\to T}{\lim}\left[2\times {\left(\frac{1}{2}\right)}^{1-\frac{t}{T}}\right]=\underset{t\to T}{\lim}{\left(\frac{1}{2}\right)}^{\frac{t}{T}}=2,\\ \begin{array}{c} \underset{t\to 0}{\lim}\left[tanh\left[2\times \left(1-\frac{t}{T}\right)\right]\right]=tanh\left(2\right),\\ \underset{t\to T}{\lim}\left[tanh\left[2\times \left(1-\frac{t}{T}\right)\right]\right]=0. \end{array} \end{array} \end{array}$$

From (8), when $t\in T_{1}\cup T_{3}$, $r_{1}^{*}\left(t\right)$ increments from 1 to 2. At this stage, the SCA will exhibit a global search. When $t\in T_{2}\cup T_{4}$, $r_{1}^{*}\left(t\right)$ decreases from $tanh\left(2\right)$ to 0. In this phase, the SCA exhibits a local search. The introduction of the multi-level adjustment factor makes the SCA switch between exploration and operation several times during the process of finding the optimum, which reduces the chances of the SCA falling into a local optimum to some extent.

Based on the conventional SCA, the logistic chaos factor is added to initialize the distribution of population solutions in order to enhance the diversity of the initial population of the optimization algorithm. The multilevel regulatory factor, $r_{1}^{*}\left(t\right),$ replaces the original $r_{1}$, in order that the SCA can change its convergence strategy according to the number of training rounds during the optimization iteration and prevent the situation of falling into a local optimum. With the addition of the logistic chaos operator and the multilevel adjustment factor strategy, the solution is updated as follows.
(9)$$\begin{array}{c} \;\;\;\;\;\;\;\;\;\;\;X_{t+1}\left(i,j\right)=\left\{\begin{array}{c} X_{t}\left(i,j\right)+r_{1}^{*}\left(t\right)sinr_{2}\left|r_{3}\times P_{t}\left(i,j\right)-X_{t}\left(i,j\right)\right|,\;r_{4}<0.5\\ X_{t}\left(i,j\right)+r_{1}^{*}\left(t\right)cosr_{2}\left|r_{3}\times P_{t}\left(i,j\right)-X_{t}\left(i,j\right)\right|,\;r_{4}\geq 0.5 \end{array}.\right. \end{array}$$

The algorithm of the MetaSCA model is shown in Algorithm 1, and the structure diagram is shown in Figure 3.
**Algorithm 1.** MetaREC process**1. Input:** Number of solution pop, dimension of solution dim, maximum number of iterations T, objective fitness function $f\left(x\right)$.
**2.**  Initialization of the initial population distribution based on pop and dim using the Logistic chaos operator.**3.**  The fitness value of each solution $X_{t}$ is calculated according to the fitness function $f\left(x\right)$ and the one $X_{t}^{best}$ with the smallest fitness value is found.**4.   Do**  (for each iteration)**5.**         Update multilevel regulatory factor $r_{1}^{*}\left(t\right)$ according to (7);**6.**         Update parameters $r_{2},r_{3,}\;r_{4}$;**7.**         The position of each solution is updated according to Equation (9);**8.**         Calculate the fitness value of each solution according to $f\left(x\right)$.**9.**         Update the global optimal solution $X_{t}^{best}$.**10.   While** (t<T)**11. Output:** Global optimal solution $X_{t}^{best}$ after iteration.

### 4.2. Deep Convolution Based LSTM Network

#### 4.2.1. Basic LSTM Process

The LSTM is a temporal recurrent neural network [63]. The LSTM is used for very long intervals of events in a time series by adding three control units, the input door, the forgotten door, and the output door [64,65,66]. The structure of the LSTM is given in Figure 4 and the main process of the LSTM is shown below.
The cellular information from the previous moment is selectively filtered using the forgotten door to pick out the cellular information that has an impact on that moment before being fed into the neural network for calculation
(10)$$\begin{array}{c} \;\;\;\;\;\;\;\;\;\;\;\;\;\;\;f_{t}=\sigma \left(w_{f}\left[h_{t-1},\;x_{t}\right]+b_{f}\right), \end{array}$$where $f_{t}$ is the output of the forgetting gate, σ is the activation function, $w_{f}$ and $b_{f}$ are the weight coefficient and offset, respectively, $h_{t-1}$ is the hidden state in the previous time series, and $x_{t}$ is the input data of the current series.
2.The input door determines which information will be stored in the cell state.
(11)$$\begin{array}{c} \;\;\;\;\;\;\;\;\;\;\;\;\;\;\;i_{t}=\sigma \left(w_{i}\left[h_{t-1},\;x_{t}\right]+b_{i}\right). \end{array}$$
(12)$$\begin{array}{c} \;\;\;\;\;\;\;\;\;\;\;\;\;\;\;\;\;\;a_{t}=\tanh\left(w_{c}\left[h_{t-1},\;x_{t}\right]+b_{c}\right). \end{array}$$
(13)$$\begin{array}{c} \;\;\;\;\;\;\;\;\;\;\;C_{t}=f_{t}\cdot C_{t-1}+i_{t}\cdot a_{t}. \end{array}$$where $i_{t}$ is the input door, $a_{t}$ is the state of the node at moment t, $C_{t-1}$ and $C_{t}$ denote the state of the cell at moments t−1 and t, respectively, and $h_{t-1}$ denotes the output at moment t−1.

3.The output gate is calculated to obtain the current hidden layer state $h_{t}$.

(14)$$\begin{array}{c} \;\;\;\;\;\;\;\;\;\;\;\;\;\;\;\;O_{t}=\sigma \left(w_{o}\left[h_{t-1},\;x_{t}\right]+b_{o}\right). \end{array}$$(15)$$h_{t}=O_{t}\cdot tanh\left(C_{t}\right).$$
where $O_{t}$ is the value obtained from the previous hidden layer state, $h_{t-1}$, together with the current layer input, $x_{t}$, after computing σ.

In addition to the parameters mentioned above, the four parameters that need to be set manually and are important for the prediction efficiency of the LSTM model are the learning rate, α, the number of training sessions, epochs, the number of neurons in the first layer, $N_{1}$, and the number of neurons in the second layer, $N_{2}$. The MetaREC will be used to find the best combination of parameters $\left(\alpha ,\;\;epochs,\;\;N_{1},\;\;N_{2}\right)$ in the LSTM model in view of its excellent merit-seeking capability.

#### 4.2.2. MetaREC Process via LSTM Network

When it comes to choosing values for the parameters $\left(\alpha ,\;\;epochs,\;\;N_{1},\;\;N_{2}\right)$ in the LSTM, the most widespread approach is to let the parameters vary within a limited range. With a set of parameters selected, the training set is applied to train the LSTM and obtain prediction accuracy. Finally, the set of parameters that obtains the best prediction accuracy is selected as the optimum parameter combination. In this experiment, the MetaREC is used to find the optimum combination of parameters. The MetaREC_LSTM obtains the optimum prediction accuracy by the following steps. The structure diagram and pseudo-code of the MetaREC_LSTM are shown in Figure 5 and Algorithm 2, respectively.

Step 1. Determine the initial parameters in the MetaREC_LSTM, such as the number of solutions, pop, the dimensionality of the solutions, dim, the upper and lower boundariess on the values of the solutions, $\left[lb,\;\;ub\right]$, and the number of iterations for the optimization search, Iternum.

Step 2. The combination of parameters $\left(\alpha ,\;\;epochs,\;\;N_{1},\;\;N_{2}\right)$ is used as the location of the solution, and the location of the initial solution is initialized using the logistic chaos operator.

Step 3. The fitness value of each solution is calculated according to (1), and the fitness of each solution also represents the training error obtained using the reformulation parameters.

Step 4. The position of each solution is updated according to (9), the fitness of each solution is recalculated, and the combination of parameters for which the minimum fitness is obtained is taken as the best combination of parameters.

Step 5. The best combination of parameters is applied to train the LSTM model and make predictions on the testing set.

The values of the important parameters in the LSTM model have a significant impact on the predictive performance of the model. Therefore, it is important to use effective methods to determine the values of these parameters.
**Algorithm 2.** MetaREC_LSTM**1. Input:** The data set, maximum number of iterations T, number of solutions pop,            dimension of solution dim, position pbest and optiaml fitness gbest of the solution X.           $X=\left[\begin{array}{c} X\left(1,\left[:dim\right]\right)\\ X\left(2,\left[:dim\right]\right)\\ \begin{array}{c} \vdots \\ X\left(pop,\left[:dim\right]\right) \end{array} \end{array}\right]$, $X\left(i,\left[:dim\right]\right)=\left[\begin{array}{ccc} X\left(i,\;1\right) & X\left(i,\;2\right) & \begin{array}{cc} X\left(i,\;3\right) & X\left(i,\;4\right) \end{array} \end{array}\right]$
**2. for** i=1 to pop **do**
**3.**    $X_{i}^{1}\to \alpha$, $X_{i}^{2}\to epochs$, $X_{i}^{3}\to N_{1}$, $X_{i}^{4}\to N_{2}$.
**4.**     / /∗∗∗∗∗∗**Calculate the fitness value of each solution by using MetaREC**∗∗∗∗∗∗ / / **5.**    LSTMtrain ($X\_train,Y\_train,\alpha ,\;\;epochs,\;\;N_{1},\;\;N_{2})\to Model$**6.**    LSTMpredice $\left(X_{test},\;Y_{test},\;Model\right)\to Result$**7.**    $Result\to fitness\left(X_{i}\right)$**8. end for****9.**∕∕∗∗∗∗∗∗**Recalculate the fitness value of each solution for updating the optimal solution according**                  **to MetaREC**∗∗∗∗∗∗ / / **10. for**t=1 to Iternum **do**
**11.**    Update parameters $r_{1},r_{2},r_{3,}\;r_{4}$;
**12.    for**i=1 to pop **do**
**13.       for j**=1 to dim **do**
**14.**          $X_{t+1}\left(i,j\right)=\left\{\begin{array}{c} X_{t}\left(i,j\right)+r_{1}^{*}\left(t\right)sinr_{2}\left|r_{3}\times P_{t}\left(i,j\right)-X_{t}\left(i,j\right)\right|,\;r_{4}<0.5\\ X_{t}\left(i,j\right)+r_{1}^{*}\left(t\right)cosr_{2}\left|r_{3}\times P_{t}\left(i,j\right)-X_{t}\left(i,j\right)\right|,\;r_{4}\geq 0.5 \end{array}.\right.$**15.       end for****16.    end for****17.**    $Minimum\;value\;of\;fitness\left(X_{t}\left(i,j\right)\right)\to X_{t}^{best}\left(i,j\right),\;gbest$**18. end for****19.**$X_{t}^{best}\left(i,1\right)\to \;\alpha$, $X_{t}^{best}\left(i,2\right)\to \;epochs$, $X_{t}^{best}\left(i,3\right)\to \;N_{1}$, $X_{t}^{best}\left(i,4\right)\to \;\;N_{2}$**20.** / /∗∗∗∗∗∗**Retraining the LSTM model**∗∗∗∗∗∗ / / **21.** LSTMtrain ($X\_train,Y\_train,\;\alpha ,\;\;epochs,\;\;N_{1},\;\;N_{2})\to Model$
**22.** LSTMpredice $\left(X\_test,\;Y\_test,\;Model\right)\to Predict$


### 4.3. LSTM-Based Heuristic Structure for Electric Forecasting

In the LSTM model, there are four parameters that play a crucial role in the prediction performance of the model, such as the number of neurons, hidden_node_1,  hidden_node_2, of the LSTM, the learning rate, alpha, and the training number, num_epochs. The values of these four parameters are taken as the object of the MetaREC optimization. The flow chart used by MetaREC_LSTM to forecast the electric load is shown in Figure 6. 

The steps for forecasting the electric load using the MetaREC_LSTM model are as follows.

Step 1: Data preprocessing. The acquired data are normalized and split into training and testing sets.

Step 2: Constructing the LSTM prediction model. Set the number of neurons, $N_{1}$ and $N_{2}$, the learning rate, α, and the range of values for the number of training iterations, epochs.

Step 3: Build the MetaREC model. Initialize the parameters of the model, including the number of populations, individual dimensions, and the maximum number of iterations, where each solution is a $\left(\alpha ,\;\;epochs,\;\;N_{1},\;\;N_{2}\right)$ combination, and the dimension of each solution is 4.

Step 4: Set the fitness function. The mean of squared errors is set as the fitness function, as shown in (16).
(16)$$\begin{array}{c} f\left(x\right)=\frac{1}{n}\overset{n}{\underset{i=1}{\sum }}{\left(y_{i}-\bar{y_{i}}\right)}^{2}. \end{array}$$

Step 5: The fitness value of each solution is calculated to determine the optimum solution in the population and its corresponding optimum fitness value.

Step 6: The position of each solution is updated according to (9), and the fitness value of each solution is recalculated to update the historical optimum value and the optimum solution.

Step 7: Terminate the iteration. Output the optimum combination of parameters $(\alpha ,\;\;epochs,\;\;N_{1},\;\;N_{2}$) and the optimum fitness at this point, when the number of iterations reaches the set maximum number of iterations.

Step 8: Forecasting of electrical loads. The best combination of parameters obtained in step 7 is used as the parameter values for the LSTM model, which is used to fit the training set and then used to make predictions on the testing set.

## 5. Simulation Results

The main configuration of the experimental platform used to evaluate the performance of all the prediction models in this paper was an Intel (R) Core (TM) i7-10750H 2.6 GHz processor, 16G memory.

The purpose of this section is to illustrate the performance of the proposed method. Considering that the PSO, WOA, and conventional SCA are more efficient search algorithms in the category of population intelligence domain. The search capability of the MetaREC proposed in this paper is compared with the PSO, WOA, and conventional SCA in the first part of the simulation results to explore the search capability and convergence speed of MetaREC. In the second part, the MetaREC is used to adjust the set parameters of the LSTM model to predict the electricity load. The prediction results are compared with the results of some other prediction models. Information about the parameters of each optimization algorithm and their initial values are given in Table 2.

### 5.1. Evaluation Setup

Different evaluation parameters for various electric prediction models in simulation experiments are given as follows:Relative percentage error: the magnitude of this parameter illustrates the difference between the predicted and true data of the load. The smaller this parameter is, the better the prediction of the model is [67].
(17)$$\begin{array}{c} \;\;\;\;\;\;\;\;\;\;\;\;\;\;\;\;\;\;\;\;\;\;error_{i}=\frac{\left|\bar{y_{i}}-y_{i}\right|}{y_{i}}\times 100\%. \end{array}$$

Mean absolute percentage error (MAPE): if this parameter is 0 the prediction model is perfect and when this parameter is greater than 100%, it means that the model is inferior [68].


(18)
$$\begin{array}{c} MAPE=\frac{1}{n}\overset{n}{\underset{i=1}{\sum }}\frac{\left|\bar{y_{i}}-y_{i}\right|}{y_{i}}. \end{array}$$


Root mean square error (RMSE): The smaller this parameter is, the better the prediction model is, and vice versa, the bigger the value the worse the model is [69].


(19)
$$\begin{array}{c} RMSE=\sqrt{\frac{1}{n}\overset{n}{\underset{i=1}{\sum }}{\left(\bar{y_{i}}-y_{i}\right)}^{2}}. \end{array}$$


Mean absolute error (MAE): The smaller this parameter is, the better the prediction performance of the prediction model.


(20)
$$\begin{array}{c} MAE=\frac{1}{n}\overset{n}{\underset{i=1}{\sum }}\left|\bar{y_{i}}-y_{i}\right|. \end{array}$$


Coefficient of determination ($R^{2})$: This parameter implies the degree of fit of the prediction model. The closer the value of this parameter is to 1, the better the fit of the model. It is the proportion of variation in the dependent variable that is predicted by the model [70].

(21)$$\begin{array}{c} R^{2}=1-\frac{\sum_{i=1}^{n}{\left(\bar{y_{i}}-y_{i}\right)}^{2}}{\sum_{i=1}^{n}{\left(y_{i}-\bar{\bar{y}}\right)}^{2}}, \end{array}.\;$$
where $y_{i}$ denotes the real power load data of the i−th time period, $\bar{y_{i}}$ denotes the forecast electric load data of the i−th time period, and $\bar{\bar{y}}$ is the mean value of the real power load data. 

### 5.2. Test Functions Assessment

The purpose of this section is to test the superiority seeking capability of the MetaREC method in solving complex optimization problems. The commonly used benchmark functions are of three types: regularity, separability, and multimodality. 

In addition, as the dimensionality of the search space increases, the difficulty of function finding increases. Therefore, six benchmark functions were selected and the search space dimensions were set to 10, 50, and 100, respectively, to test the efficient search capability of the MetaREC. At the same time, the results obtained by the MetaREC were compared with those acquired by PSO, WOA, and SCA. The comparison results are shown in Table 3 and Figure 7, Figure 8 and Figure 9. The expressions of the six benchmark functions are as follows:(22)$$\begin{array}{c} Sphere\;:f\left(x\right)=\overset{dim}{\underset{i=1}{\sum }}x_{i}^{2}, \end{array}$$
(23)$$\begin{array}{c} Rastrigin\;:\;f\left(x\right)=\overset{dim}{\underset{i=1}{\sum }}\left[x_{i}^{2}-10cos\left(2\pi x_{i}\right)+10\right], \end{array}$$
(24)$$\begin{array}{c} Quartic\;:\;f\left(x\right)=\overset{dim}{\underset{i=1}{\sum }}ix_{i}^{4}+random\left[0,\;1\right]\;,\left(24\right) \end{array}$$
(25)$$\begin{array}{c} Griwank\;:\;f\left(x\right)=\overset{dim}{\underset{i=1}{\sum }}\frac{x_{i}^{2}}{4000}-\overset{dim}{\underset{i=1}{\prod }}cos\left(\frac{x_{i}}{\sqrt{i}}\right)+1\;, \end{array}$$
(26)$$Ackley\;:\;f\left(x\right)\;=-20exp\left(-0.2\sqrt{\frac{1}{dim}\overset{dim}{\underset{i=1}{\sum }}x_{i}^{2}}\right)-exp\left(\frac{1}{dim}\overset{dim}{\underset{i=1}{\sum }}cos\left(2\pi x_{i}\right)\right)\;+20+e,$$
(27)$$\begin{array}{c} Step\;:\;f\left(x\right)=\overset{dim}{\underset{i=1}{\sum }}{\left(x_{i}+0.5\right)}^{2}. \end{array}$$

Each algorithm was run 10 times to obtain the mean and variance of the algorithm’s search results. In Table 3, the results of the five optimization algorithms (SCA, PSO, FA, WOA, and MetaREC) are shown in different dimensions for the benchmark function search. For the sphere function, the MetaREC obtained better optimum values and variances in all three dimensions than the results of the other four methods. For the Rastrigin function, although the conventional SCA can reach the theoretical optimum at dim = 10, it can be obtained from the results in the table that as the dimensionality of the search space increases, the SCA’s ability to find the optimum decreases, yet the MetaREC can always obtain the theoretical optimum. For the quartic, groan, and step functions, the optimum values and variances obtained by the MetaREC are better than those obtained by the SCA, PSO, and FA. Only the results obtained by the WOA are similar with those of the MetaREC. In addition, the MetaREC obtained the theoretical optimum value in the search for the optimal values for the Griwank and step functions. The above simulation experimental results and analysis show that the MetaREC mentioned in this paper has considerable accuracy and stability in dealing with low, medium, and high dimensional problems.

With the purpose of showing the convergence speed of MetaREC during the optimization, this paper shows the iteration curves of five optimization algorithms for optimization on six benchmark functions at dim=10, dim=50, and dim=100, respectively.

Firstly, the speed of convergence of the different optimization algorithms on the six benchmark functions is tested by setting dim=10. The test results are shown in Figure 7. As shown in Figure 7a,b,d–f, the MetaREC not only gives more accurate iterative results, but also has the fastest convergence rate among the five optimization algorithms. Although the convergence accuracy of the MetaREC is slightly lower than that of WOA in Figure 8c, the convergence speed is significantly higher than that of WOA.

Secondly, dim=50 was set to test the results of iterations of the five algorithms on the six benchmark functions. The simulation results are exhibited in Figure 8. In Figure 9a, MetaREC achieves the optimum accuracy along with the optimum iterative convergence speed. As shown in Figure 8b,d–f, the convergence speed and accuracy of the MetaREC’s optimization search are significantly higher than the results of the SCA, PSO, and FA. Although the convergence accuracy of WOA can be the same as that of the MetaREC, the convergence speed of the MetaREC is significantly greater than that of the WOA.

Finally, dim=100 was set to test the convergence results of the different algorithms on the six benchmark functions. The simulation results are presented in Figure 9, the convergence speed of the MetaREC almost does not change as the dimensionality of the search space increases. In Figure 9b,d–f, the convergence speed of the MetaREC is still the fastest among the five algorithms. From the above experimental results it can be obtained that in most cases the MetaREC can guarantee higher accuracy while also obtaining faster convergence.

### 5.3. Comparison of Electricity Load Forecasting

The purpose of this section is to use the forecasting techniques in this paper to forecast electrical loads and to test the performance of the forecasting techniques. In this experiment, published electrical load data from a region in Zhejiang, China, is applied. This dataset includes the electrical load values for the period 0–23 h for each day from 13 February to 20 May 2021. The electricity load data from 13 February to 19 May were used as the training set to train the forecasting model and the data from 20 May were used as the testing set to check the performance of the forecasting technique. To verify the efficiency of MetaREC_LSTM technology, the test results are compared with those of LSTM, WOA_LSTM, and SCA_LSTM models. 

Table 4 gives the absolute values of the deviations between the predicted and actual values of the electric load for the 0–23-h period using the three forecasting models—WOA_LSTM, SCA_LSTM, and MetaREC_LSTM. 

Firstly, it can be obtained from Table 4 that the minimum and maximum absolute values of the deviations in the forecasts obtained by WOA_LSTM are 0.01% at the 21:00 moment and 4.79% at the 6:00 moment, respectively. The minimum and maximum absolute values of the deviations in the forecasts obtained by SCA_LSTM are 0.53% at 16:00 and 4.82% at 6:00, respectively. The minimum and maximum absolute values of the deviations in the forecasts obtained by MetaREC_LSTM are 0.02% at 0:00 and 3.18% at 6:00, respectively. From the above resulting data, it can be obtained that the minimum deviation of MetaREC_LSTM prediction results are similar to the minimum deviation obtained by WOA_LSTM and is smaller to the minimum deviation obtained by SCA_LSTM. Furthermore, the maximum deviation obtained by MetaREC_LSTM is significantly better than the maximum deviation obtained by WOA_LSTM and SCA_LSTM. Secondly, the range $\left[-2\%,\;2\%\right]$ was chosen to observe the number of predicted values within this range for each prediction model. Both the WOA_LSTM model and the SCA_LSTM model had 16 values in this range, while the MetaREC_LSTM model had 17. In order to show graphically, the comparison of the errors of each prediction model are shown in Figure 10. It is observed that the line of errors at each time point obtained from the MetaREC_LSTM model is closer to Y=0 than the line of prediction errors from the LSTM, WOA_LSTM, and MetaREC_LSTM models, again indicating a higher stability and accuracy of the MetaREC_LSTM prediction model. 

Using the error data from Table 4, the minimum and maximum of absolute values of errors were found, the first quartile, the median error and the third quartile, mean and the interquartile range for each method. 

From the Table 5 and Figure 11, the minimum error, median, and mean of WOA_LSTM and MetaREC_LSTM are quite close but the upper 50% of error of the WOA_LSTM spans far wider than for the MetaREC_LSTM. For instance, the third quartile is more than 20% bigger for the WOA_LSTM, and the maximum value is 50.6% bigger than those for the MetaREC_LSTM. The MetaREC_LSTM has slightly bigger mean and median for the first quartile than those for the WOA_LSTM and somewhat smaller than the SCA_LSTM for these parameters. The MetaREC_LSTM has the third quartile almost the same as that of the SCA_LSTM. The absolute value of the maximum error of the SCA is almost the same as that of the WOA and considerably bigger than that of the MetaREC. Thus, the method considerably reduces the magnitude of prediction errors. 

In Table 6, the values of MAPE, RMSE, MAE, and $R^{2}$ are given for the BP, LSTM, WOA_LSTM, SCA_LSTM, and MetaREC_LSTM for the testing dataset. From Table 6, the values of MAPE, RMSE, MAE, and $R^{2}$ obtained by the MetaREC_LSTM prediction technique proposed in this paper are the best among the five prediction methods mentioned above. The MetaREC_LSTM method predicts approximately 28%, 33%, and 29% lower than the LSTM results for MAPE, RMSE, and MAE, respectively; 5%, 15%, and 9% lower than the WOA_LSTM results, respectively; and 23%, 24%, and 25% lower than the SCA_LSTM results, respectively. In the comparison of $R^{2}$ as an indicator, MetaREC is approximately 1.00%, 0.31%, and 0.59% higher than LSTM, WOA_LSTM, and SCA_LSTM, respectively.

The graphic presentation is given in Figure 12. It shows the predicted load curves obtained by the five prediction methods and true load curves. As can be seen in Figure 12, compared to the other four forecasting methods, the load forecasting curve of MetaREC_LSTM fits better with the real load curve, indicating that MetaREC_LSTM has higher forecasting accuracy and proving the efficiency of MetaREC_LSTM in electricity load forecasting.

In order to make the conclusions obtained more general, different testing sets and training sets were used to fit and test the prediction methods. Electricity load data from 13 February to 18 May were used as the training set and load data from 19 May were used as the testing set. 

The error rate curves obtained by the four prediction methods of the LSTM, WOA_LSTM, SCA_LSTM, and MetaREC_LSTM were compared for each time point and the results are shown in Figure 13. The MetaREC_LSTM achieves better error rates than the other three forecasting methods for electricity forecasting at most points in time. Figure 14 shows the load forecasting curves of different forecasting methods on 19 May, and it can be obtained that the forecasting curves of this paper’s method fit the real load data curve more closely than other methods in most of the time periods. From the above data and analysis, it can be confirmed that the MetaREC_LSTM has higher forecasting performance.

The values of MAPE, RMSE, MAE, and $R^{2}$ for each forecasting method on 19 May for the training set were compared and the results exhibited in Figure 15 and Table 7. It can be clearly seen that the MAPE, RMSE, and MAE data obtained by the MetaREC_LSTM are significantly smaller than for the other four methods and are equivalent to the results of WOA_LSTM and SCA_LSTM in terms of $R^{2}$. In order to show more clearly the comparison of data on the assessment indicators for the five forecasting methods, Table 7 displays the exact data obtained for the five methods for the abovementioned indicators. From Table 7, the MAPEs obtained by MetaREC_LSTM were 54%, 52%, 7%, and 5% lower than the results obtained by BP, LSTM, WOA_LSTM, and SCA_LSTM, respectively. The RMSEs are 49%, 46%, 2.8%, and 3.2% lower than the results obtained by BP, LSTM, WOA_LSTM, and SCA_LSTM, respectively. The MAEs obtained are 55%, 55%, 12%, and 3.6% lower than the results obtained by BP, LSTM, WOA_LSTM, and SCA_LSTM, respectively. In addition, MetaREC_LSTM obtained improvements in $R^{2}$ of 4.8%, 4.0%, 0.04%, and 0.09% over the results of BP, LSTM, WOA_LSTM, and SCA_LSTM, respectively. The above data results and analysis can also confirm the efficient forecasting capability of the proposed MetaREC_LSTM for electricity load forecasting. 

## 6. Conclusions

Short-term power load forecasting is an important part of grid management and the foundation for power dispatch centers to develop generation plans, which has a significant responsibility in the efficient operation of power systems. Therefore, a hybrid method (MetaREC_LSTM) forecasting framework was proposed in order to improve short-term power load forecasting accuracy. The logistic chaos operator and multilevel modulation factor are used to improve the SCA. Then the improved SCA is used to optimize the parameter taken from the LSTM. Finally, the MetaREC_LSTM forecasting framework is used to forecast the electric load while comparing it with a few other single and hybrid forecasting models with respect to forecasting performance. The experimental results and analysis verify that the prediction model in this paper has higher forecast accuracy and stability. In future work, factors such as temperature, humidity, and holidays can be taken into account to improve the accuracy of the prediction model more effectively.

Due to the excellent forecasting performance of MetaREC_LSTM and its important feature of reducing the magnitude of the forecast error, it is suggested that power companies may consider applying it to their own short-term electric load forecasting for the purpose of scheduling the total amount of power generation planned by the company and thus improve the economic value of the company. Meanwhile, the prediction framework can also be applied to other prediction fields, such as wind prediction, traffic flow prediction, flu prediction, pollution prediction of complex ecosystems, etc.

## Figures and Tables

**Figure 1 sensors-22-07900-f001:**
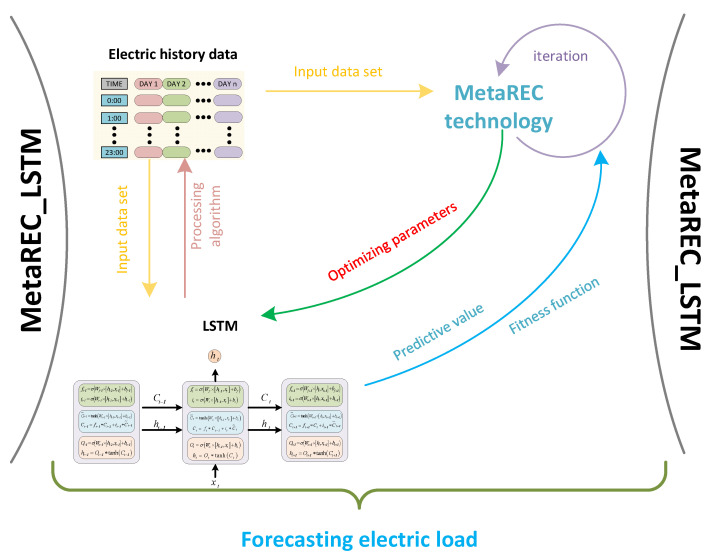
MetaREC_LSTM prediction framework for demand prediction with electricity management.

**Figure 2 sensors-22-07900-f002:**
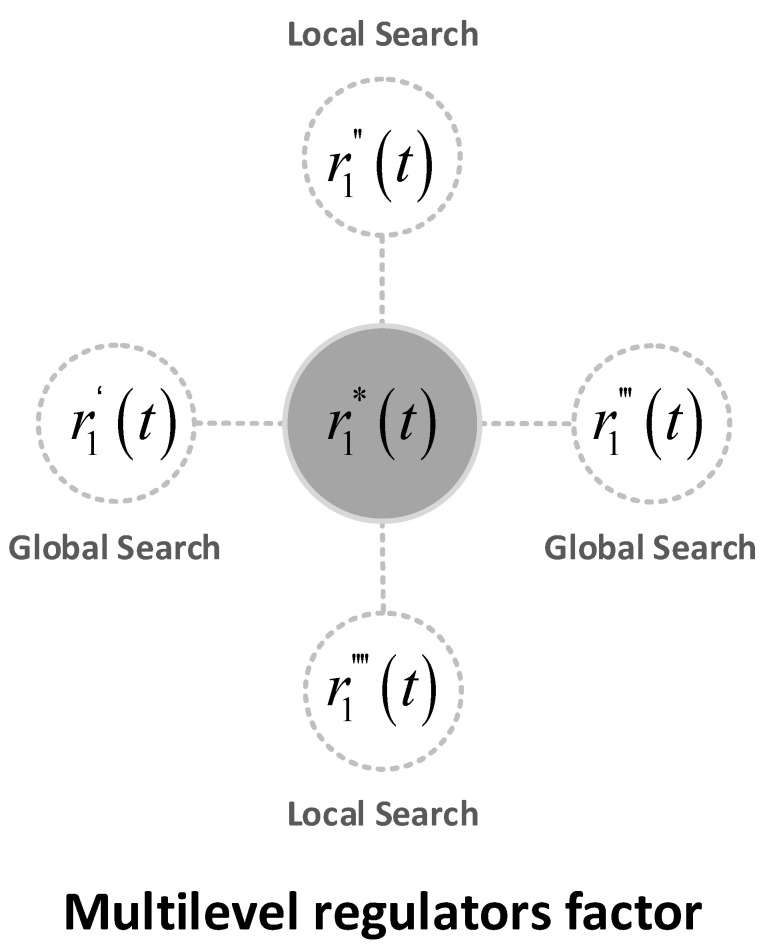
Multilevel regulatory factor.

**Figure 3 sensors-22-07900-f003:**
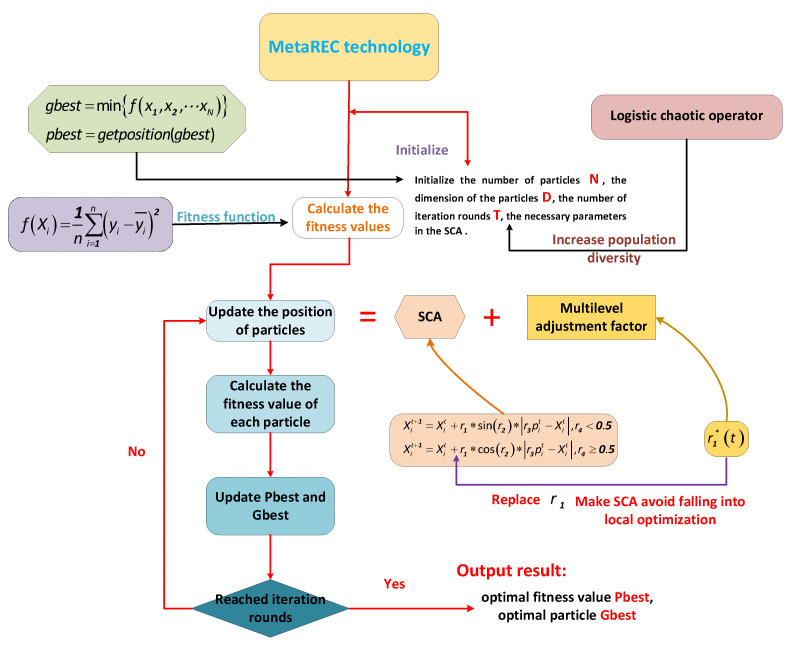
The MetaREC procedure.

**Figure 4 sensors-22-07900-f004:**
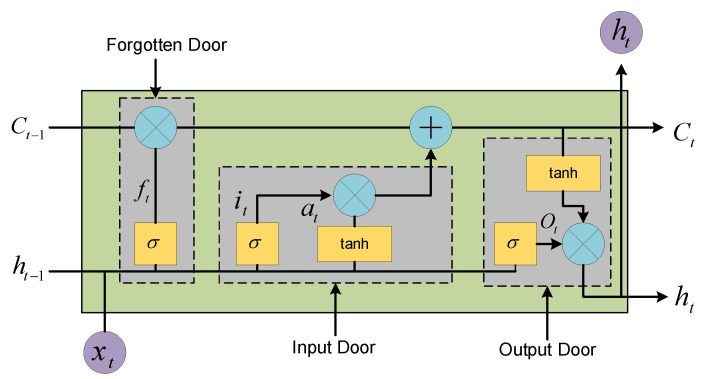
The LSTM structure diagram.

**Figure 5 sensors-22-07900-f005:**
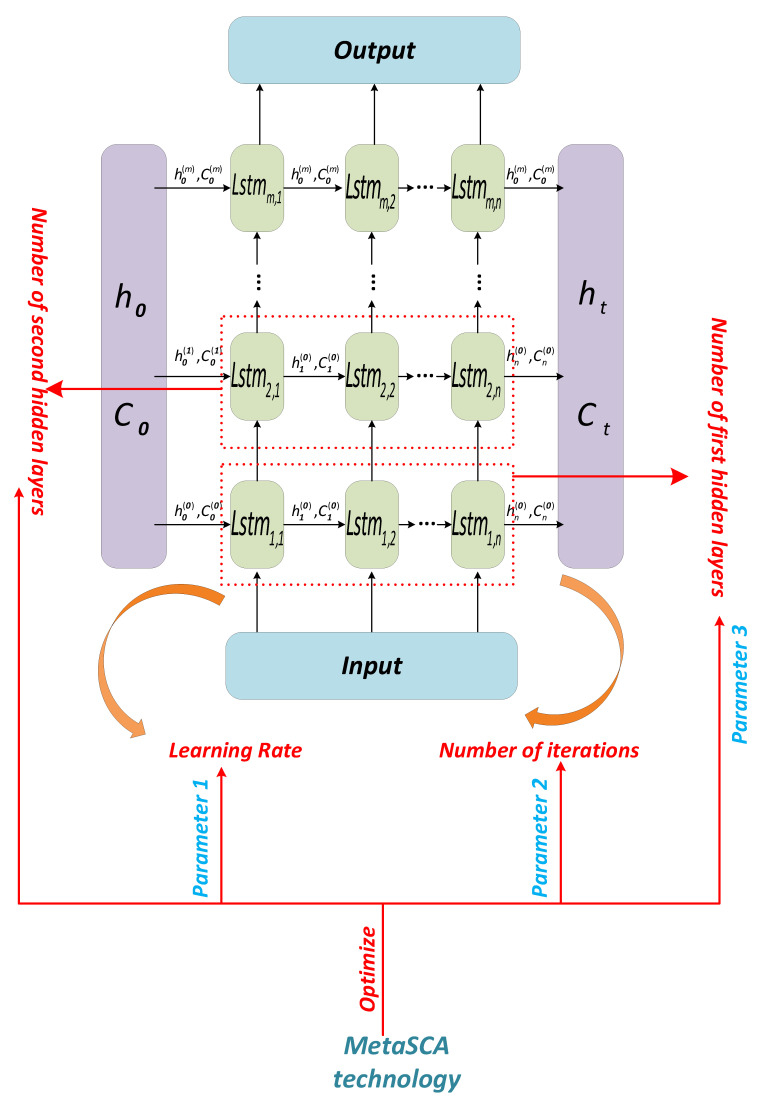
MetaREC_LSTM structure.

**Figure 6 sensors-22-07900-f006:**
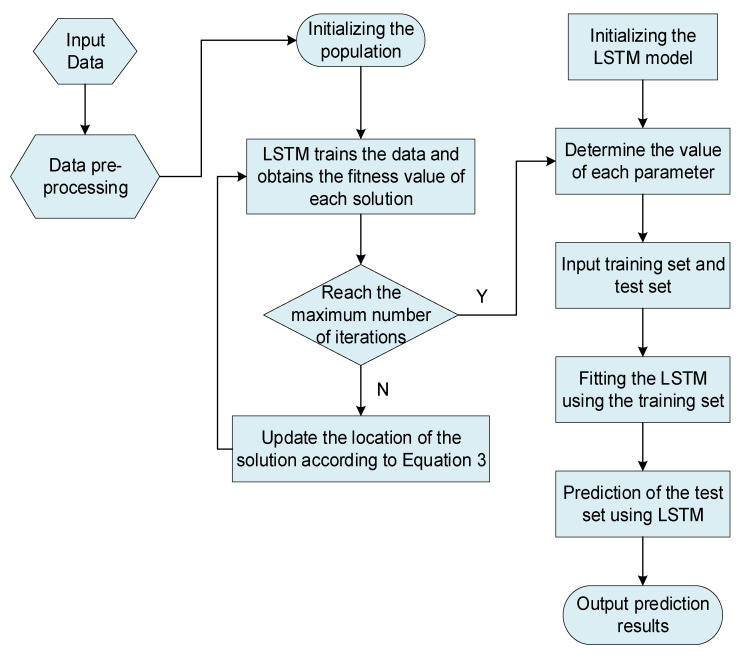
The MetaREC_LSTM solves the problem of electric load forecasting.

**Figure 7 sensors-22-07900-f007:**
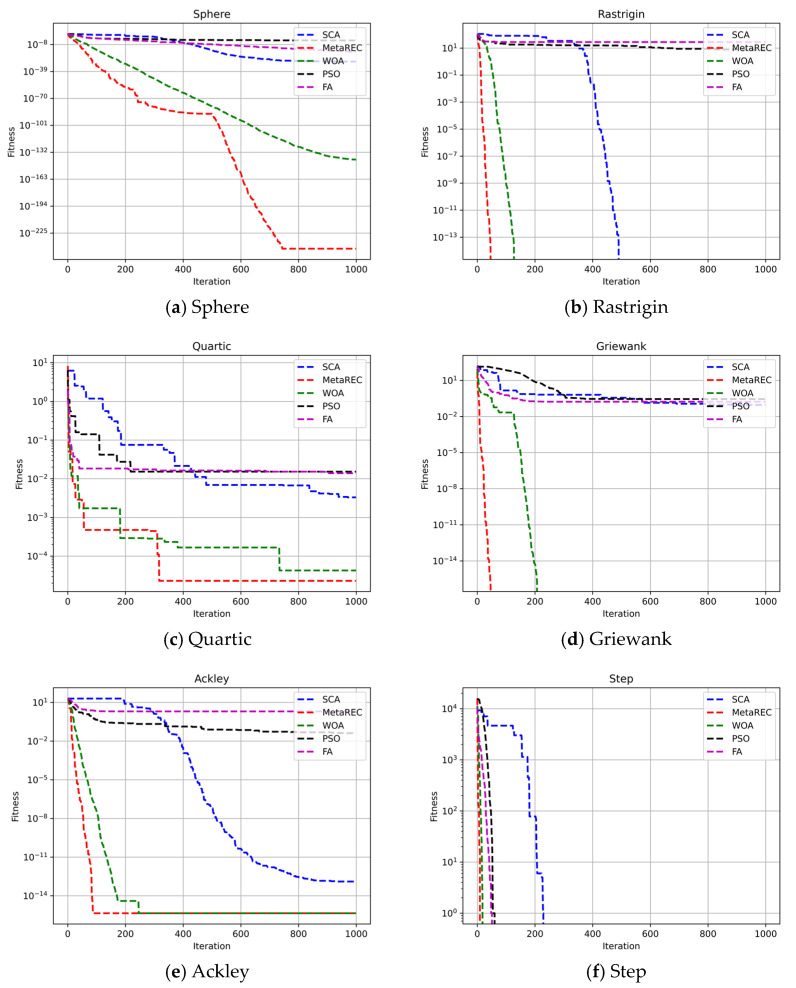
Comparison of the convergence of the SCA, PSO, FA, WOA, and MetaREC on test functions. $\left(D=10\right)$.

**Figure 8 sensors-22-07900-f008:**
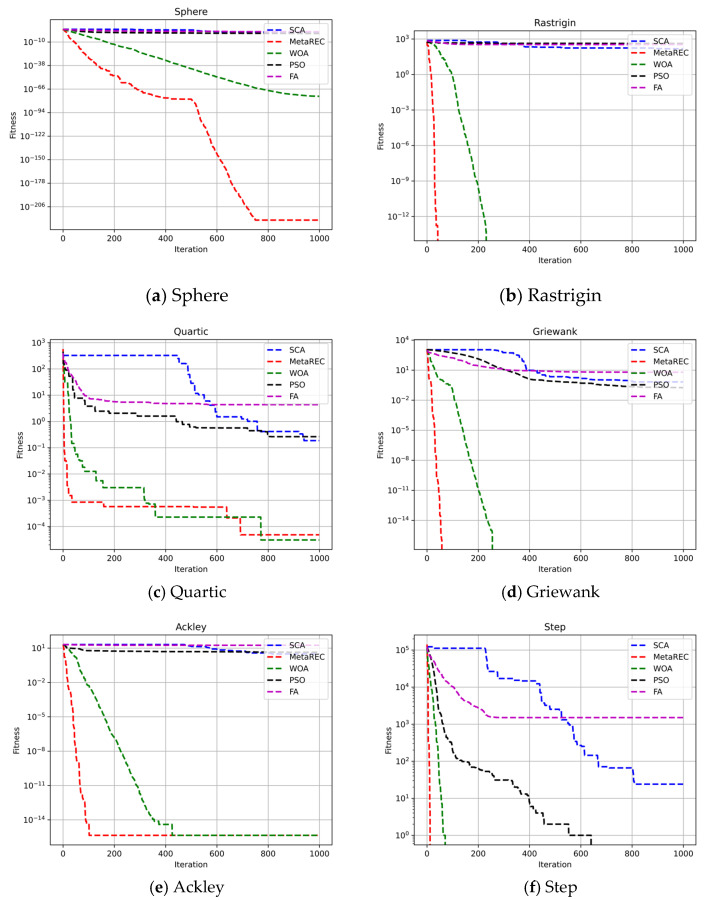
Comparison of the convergence of the SCA, PSO, FA, WOA, and MetaREC on test functions. $\left(D=50\right)$.

**Figure 9 sensors-22-07900-f009:**
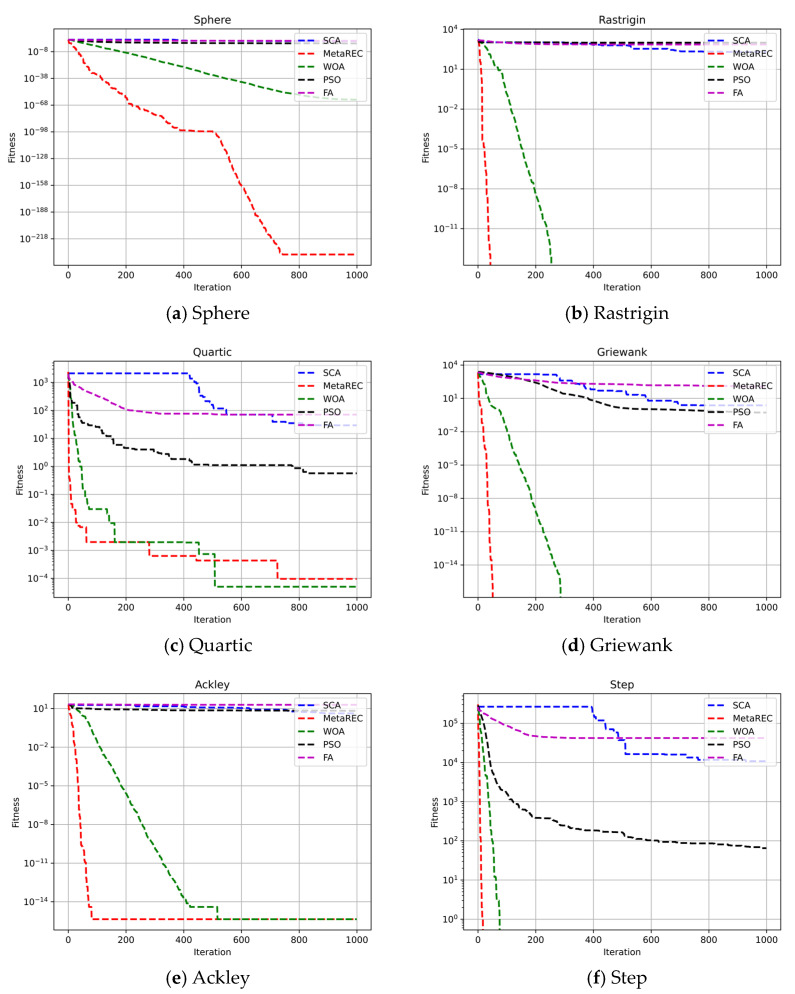
Comparison of the convergence of the SCA, PSO, FA, WOA, and MetaREC on test functions. $\left(D=100\right)$.

**Figure 10 sensors-22-07900-f010:**
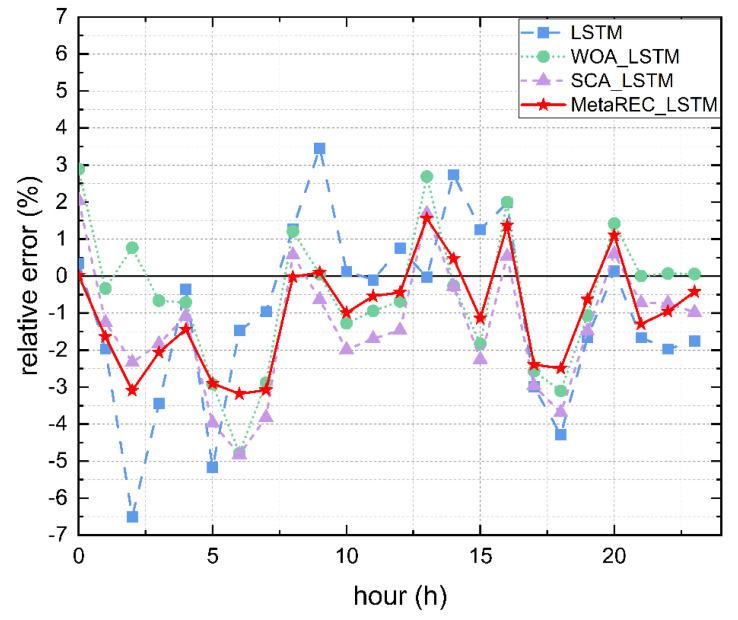
Error rate curves.

**Figure 11 sensors-22-07900-f011:**
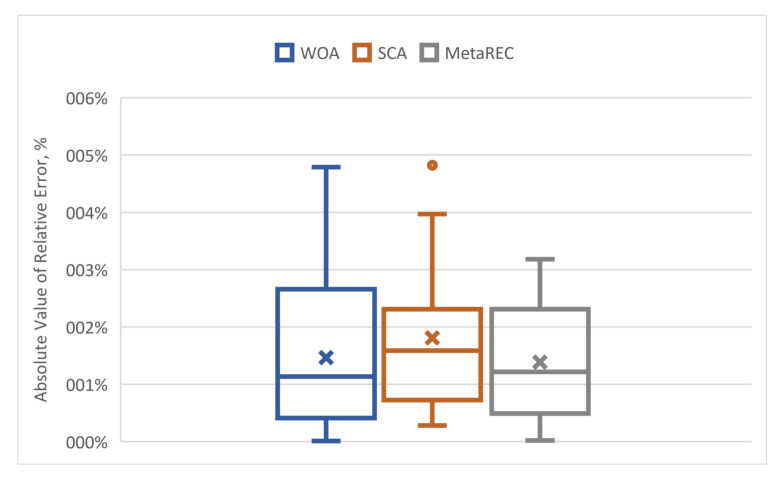
Minimum and maximum errors, mean (marked with cross), the first quartile, the median error, and the third quartile of absolute values of errors for the three methods.

**Figure 12 sensors-22-07900-f012:**
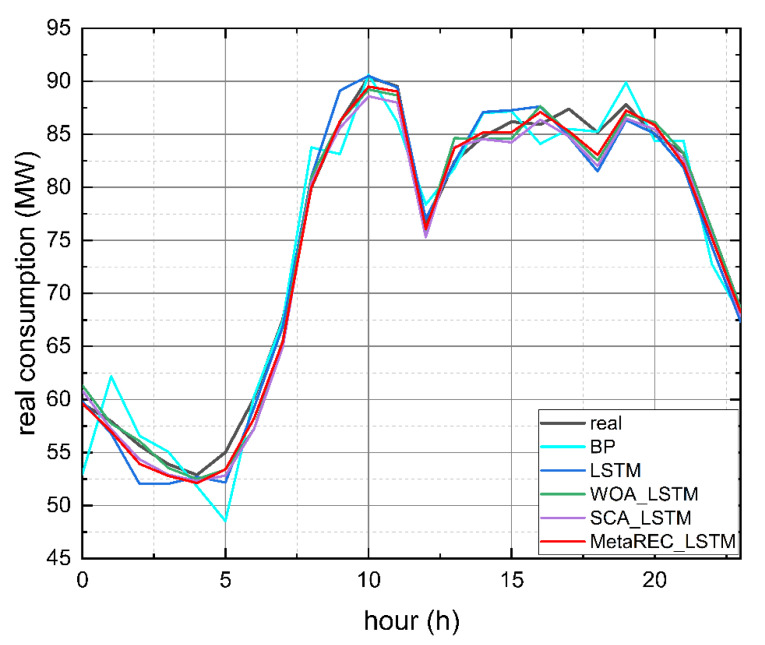
Electric load measurements and predictions made by various models.

**Figure 13 sensors-22-07900-f013:**
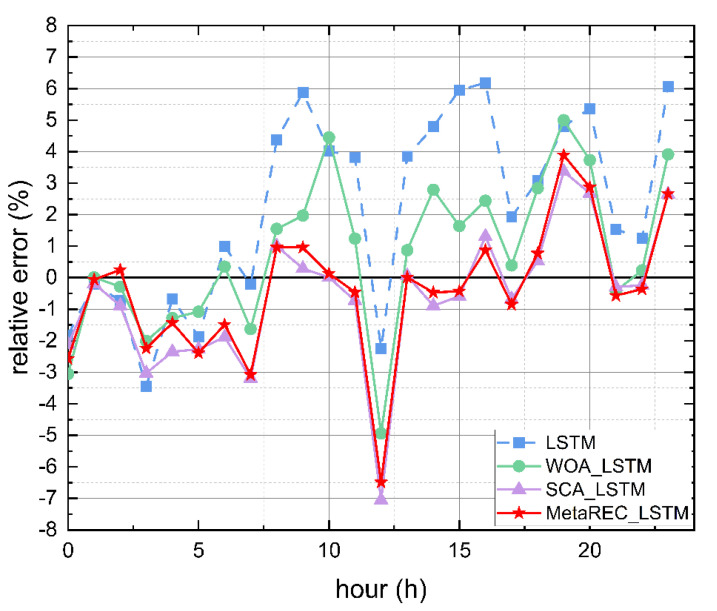
Error rate curves.

**Figure 14 sensors-22-07900-f014:**
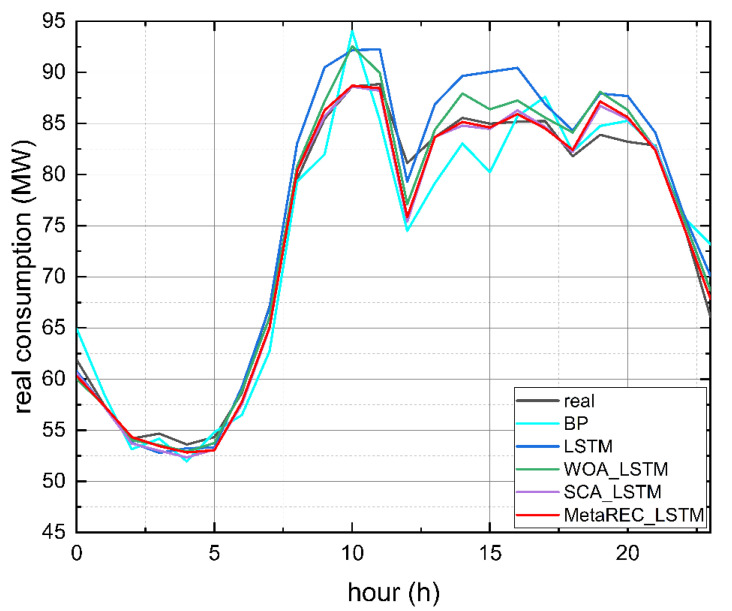
Electric load measurements and predictions made by various methods.

**Figure 15 sensors-22-07900-f015:**
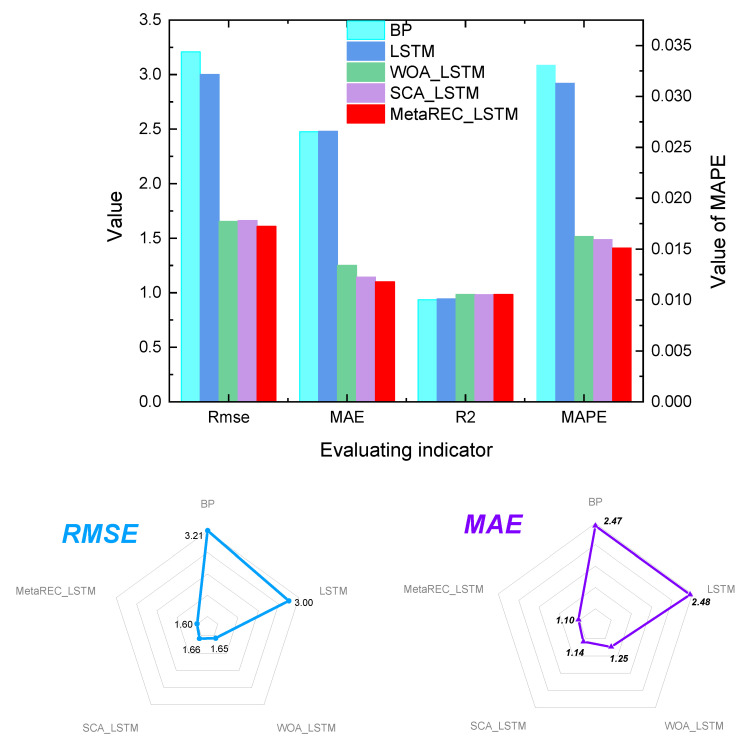
Comparison of assessment indicators for multiple forecasting methods.

**Table 1 sensors-22-07900-t001:** Symbols and their definitions.

Symbol	Definition	Symbol	Definition
$X_{t}\left(i,j\right)$	Position of solution i in the j−th dimension at the th iteration	α	Learning rate in LSTM
$P_{t}\left(i,j\right)$	Position of the optimum solution in the j−th dimension at the t−th round of iteration	epochs	Training times in the LSTM
$X_{t}^{best}$	Position of the optimum individual in the iteration	$N_{1}$	Number of nodes in the first hidden layer of the LSTM
pop	Number of solutions	$N_{2}$	Number of nodes in the second hidden layer of the LSTM
dim	Number of dimensions of the solution	$\left[Lb,\;Ub\right]$	Maximum and minimum values of the power forecast.
T	Total number of iterations of the SCA	$y_{i}$	Actual value of electric load
t	Number of current iteration rounds	$\bar{y_{i}}$	Forecasted value of electric load
$r_{1}\left(t\right)$	Regulatory factor	error	Relative percentage error of the MetaREC
$r_{2}$	Rdom Factor	MAPE	Mean absolute percentage error
$r_{3}$	Random Factor	RMSE	Root mean square error
$r_{4}$	Random Factor	MAE	Mean absolute error
$r_{1}^{*}\left(t\right)$	Multilevel regulatory factor	$R^{2}$	Coefficient of determination of the MetaREC

**Table 2 sensors-22-07900-t002:** The test function used in the experiment and some details.

No.	Function	Range	Global Optimal Value
1	Sphere	$\left[-100,100\right]$	$F_{min}=0$
2	Rastrigin	$\left[-5.12,5.12\right]$	$F_{min}=0$
3	Quartic	$\left[-1.28,1.28\right]$	$F_{min}=0$
4	Griewank	$\left[-600,600\right]$	$F_{min}=0$
5	Ackley	$\left[-32,32\right]$	$F_{min}=0$
6	Step	$\left[-100,100\right]$	$F_{min}=0$

**Table 3 sensors-22-07900-t003:** Means and variances of the SCA, PSO, FA, WOA, and MetaREC on test functions.

No.	Dim	SCA		PSO		FA		WOA		MetaREC	
		Mean	Var	Mean	Var	Mean	Var	Mean	Var	Mean	Var
1		$1.59\;\times \;10^{-28}$	$4.38\;\times \;10^{-56}$	$2.29\;\times \;10^{-3}$	$1.46\;\times \;10^{-6}$	$2.46\;\times \;10^{-17}$.	$2.86\;\times \;10^{-35}$	$2.29\;\times \;10^{-142}$	$2.01\;\times \;10^{-283}$	$3.21\;\times \;10^{-235}$	$0.00\;\times \;10^{0}$
	50	$2.22\;\times \;10^{2}$	$8.90\;\times \;10^{4}$	$1.16\;\times \;10^{0}$	$8.93\;\times \;10^{-2}$	$3.04\;\times \;10^{2}$	$1.47\;\times \;10^{4}$	$9.95\;\times \;10^{-72}$	$3.01\;\times \;10^{-142}$	$5.73\;\times \;10^{-216}$	$0.00\;\times \;10^{0}$
	100	$6.08\;\times \;10^{3}$	$3.56\;\times \;10^{6}$	1.38 × 101	$6.44\;\times \;10^{0}$	$2.10\;\times \;10^{4}$	$1.36\;\times \;10^{7}$	$6.59\;\times \;10^{-62}$	$6.01\;\times \;10^{-64}$	$7.31\;\times \;10^{-220}$	$0.00\;\times \;10^{0}$
2	10	$0\;\times \;10^{0}$	$0\;\times \;10^{0}$	$7.31\;\times \;10^{0}$	$4.94\;\times \;10^{0}$	$2.58\;\times \;10^{1}$	$2.34\;\times \;10^{2}$	$0.00\;\times \;10^{0}$	$0.00\;\times \;10^{0}$	$0.00\;\times \;10^{0}$	$0.00\;\times \;10^{0}$
		$5.37\;\times \;10^{1}$	$5.97\;\times \;10^{2}$.	$3.30\;\times \;10^{2}$	$1.58\;\times \;10^{4}$	$3.26\;\times \;10^{2}$	$5.27\;\times \;10^{3}$	$0.00\;\times \;10^{0}$	$0.00\;\times \;10^{0}$	$0.00\;\times \;10^{0}$	$0.00\;\times \;10^{0}$
	100	$1.98\;\times \;10^{2}$	$2.63\;\times \;10^{3}$	$8.76\;\times \;10^{2}$	$3.13\;\times \;10^{2}$	$6.58\;\times \;10^{2}$	$8.68\;\times \;10^{3}$	$0.00\;\times \;10^{0}$	$0.00\;\times \;10^{0}$	$0.00\;\times \;10^{0}$	$0.00\;\times \;10^{0}$
3	10	$1.72\;\times \;10^{-3}$	$2.34\;\times \;10^{-7}$	$1.52\;\times \;10^{-2}$	$6.34\;\times \;10^{-5}$	$3.69\;\times \;10^{-2}$	$1.56\;\times \;10^{-3}$	$1.73\;\times \;10^{-5}$	$2.50\;\times \;10^{-10}$	$3.14\;\times \;10^{-5}$	$6.31\;\times \;10^{-6}$
	50	$8.81\;\times \;10^{-1}$	$3.03\;\times \;10^{-1}$	$1.62\;\times \;10^{-1}$	$3.81\;\times \;10^{-3}$	$5.01\;\times \;10^{0}$	$1.56\;\times \;10^{0}$	$4.19\;\times \;10^{-5}$	$1.13\;\times \;10^{-9}$	1.02 × 10−4	$4.48\;\times \;10^{-9}$
	100	$7.60\;\times \;10^{1}$	$2.19\;\times \;10^{3}$	$5.38\;\times \;10^{-1}$	$1.31\;\times \;10^{-1}$	$8.28\;\times \;10^{-1}$	$2.71\;\times \;10^{2}$	$7.70\;\times \;10^{-5}$	$2.81\;\times \;10^{-9}$	6.88 × 10−5	$4.46\;\times \;10^{-9}$
4	10	$9.85\;\times \;10^{-3}$	$2.54\;\times \;10^{-4}$	$7.45\;\times \;10^{-1}$	$2.26\;\times \;10^{-1}$	$8.41\;\times \;10^{-2}$	$5.53\;\times \;10^{-4}$	$0.00\;\times \;10^{0}$	$0.00\;\times \;10^{0}$	$0.00\;\times \;10^{0}$	$0.00\;\times \;10^{0}$
	50	$2.05\;\times \;10^{0}$	$4.20\;\times \;10^{0}$	$1.70\;\times \;10^{-1}$	$5.34\;\times \;10^{-4}$	$4.27\;\times \;10^{0}$	$1.32\;\times \;10^{1}$	$0.00\;\times \;10^{0}$	$0.00\;\times \;10^{0}$	$0.00\;\times \;10^{0}$	$0.00\;\times \;10^{0}$
	100	$3.32\;\times \;10^{1}$	$5.34\;\times \;10^{2}$	$6.00\;\times \;10^{-1}$	$2.73\;\times \;10^{-3}$	$1.64\;\times \;10^{2}$	$1.04\;\times \;10^{3}$	$0.00\;\times \;10^{0}$	$0.00\;\times \;10^{0}$	$0.00\;\times \;10^{0}$	$0.00\;\times \;10^{0}$
5	10	$3.99\;\times \;10^{-15}$	$1.62\;\times \;10^{-8}$	$3.47\;\times \;10^{-2}$.	$2.81\;\times \;10^{-1}$	$18\;\times \;10^{-9}$	$2.68\;\times \;10^{-2}$	$4.44\;\times \;10^{-16}$	$0.00\;\times \;10^{0}$	$4.44\;\times \;10^{-16}$	$0.00\;\times \;10^{0}$
	50	$2.73\;\times \;10^{-1}$	$2.35\;\times \;10^{-1}$	$3.34\;\times \;10^{0}$	$1.72\;\times \;10^{0}$	$1.78\;\times \;10^{1}$	$1.24\;\times \;10^{2}$	$4.44\;\times \;10^{-16}$	$0.00\;\times \;10^{0}$	$4.44\;\times \;10^{-16}$	$0.00\;\times \;10^{0}$
	100	$2.06\;\times \;10^{1}$	$3.42\;\times \;10^{3}$	$5.57\;\times \;10^{0}$	$4.26\;\times \;10^{+3}$	$1.78\;\times \;10^{1}$	$2.43\;\times \;10^{3}$	$4.44\;\times \;10^{-16}$	$0.00\;\times \;10^{0}$	$4.44\;\times \;10^{-16}$	$0.00\;\times \;10^{0}$
6	10	$0.00\;\times \;10^{0}$	$0.00\;\times \;10^{0}$	$0.00\;\times \;10^{0}$	$0.00\;\times \;10^{0}$	$0.00\;\times \;10^{0}$	$0.00\;\times \;10^{0}$	$0.00\;\times \;10^{0}$	$0.00\;\times \;10^{0}$	$0.00\;\times \;10^{0}$	$0.00\;\times \;10^{0}$
	50	$4.66\;\times \;10^{1}$	$6.63\;\times \;10^{2}$	$3.40\;\times \;10^{0}$	$3.44\;\times \;10^{0}$	$2.81\;\times \;10^{4}$	$6.03\;\times \;10^{5}$	$0.00\;\times \;10^{0}$	$0.00\;\times \;10^{0}$	$0.00\;\times \;10^{0}$	$0.00\;\times \;10^{0}$
	100	$3.99\;\times \;10^{3}$	$7.41\;\times \;10^{7}$	$6.64\;\times \;10^{1}$	$7.09\;\times \;10^{2}$	$3.44\;\times \;10^{4}$	$1.18\;\times \;10^{8}$	$0.00\;\times \;10^{0}$	$0.00\;\times \;10^{0}$	$0.00\;\times \;10^{0}$	$0.00\;\times \;10^{0}$

**Table 4 sensors-22-07900-t004:** Comparison of results from multiple prediction models.

Time	Actual Load (MW)	WOA_LSTM		SCA_LSTM		MetaREC_LSTM	
		Forecast (MW)	Error	Forecast (MW)	Error	Forecast (MW)	Error
0:00	59.60	61.32	2.89%	60.83	2.05%	59.61	0.02%
1:00	57.94	57.74	–0.33%	57.21	–1.25%	56.99	–1.64%
2:00	55.65	56.08	0.77%	54.35	2.33%	53.93	–3.09%
3:00	53.90	53.54	–0.66%	52.91	–1.83%	52.79	–2.06%
4:00	52.88	52.5	–0.71%	52.31	–1.08%	52.12	–1.44%
5:00	55.01	53.39	–2.92%	52.83	–3.97%	53.41	–2.90%
6:00	60.13	57.24	–4.79%	57.23	–4.82%	58.21	–3.18%
7:00	67.51	65.56	–2.88%	64.93	–3.82%	65.43	–3.08%
8:00	80.01	80.98	1.21%	80.47	0.57%	80.00	–0.02%
9:00	86.15	86.19	0.05%	85.61	–0.63%	86.24	0.10%
10:00	90.39	89.23	–1.27%	88.59	–1.99%	89.49	–0.99%
11:00	89.52	88.67	–0.94%	88.00	–1.69%	89.04	–0.54%
12:00	76.43	75.90	–0.68%	75.31	–1.46%	76.09	–0.44%
13:00	82.42	84.64	2.69%	83.81	1.69%	83.71	1.56%
14:00	84.78	84.55	–0.26%	84.54	–0.28%	85.18	0.47%
15:00	86.19	84.61	–1.83%	84.25	–2.25%	85.20	–1.14%
16:00	85.93	87.65	2.00%	86.39	0.53%	87.11	1.37%
17:00	87.39	85.14	–2.57%	84.80	–2.96%	85.30	–2.39%
18:00	85.18	82.53	–3.10%	82.03	–3.69%	83.06	–2.49%
19:00	87.8	86.87	–1.06%	86.50	–1.48%	87.25	–0.63%
20:00	84.94	86.15	–1.42%	85.46	0.60%	85.88	1.11%
21:00	83.26	83.27	0.01%	82.66	–0.72%	82.18	–1.29%
22:00	75.92	75.98	0.07%	75.37	–0.73%	75.20	–0.95%
23:00	68.54	68.58	0.06%	67.87	–0.98%	68.25	–0.42%

**Table 5 sensors-22-07900-t005:** Comparison of the results of the three methods.

	WOA_LSTM	SCA_LSTM	MetaREC_LSTM
Minimum Value	0.01%	0.28%	0.02%
First Quartile	0.58%	0.73%	0.52%
Median	1.14%	1.59%	1.22%
Third Quartile	2.60%	2.27%	2.14%
Maximum Value	4.79%	4.82%	3.18%
Interquartile Range	2.02%	1.54%	1.62%
Mean Absolute Error	1.47%	1.81%	1.39%

**Table 6 sensors-22-07900-t006:** Results of forecasting criteria evaluated by different methods.

Method	MAPE	RMSE	MAE	$R^{2}$
BP	0.02900	2.6991	2.0110	0.9577
LSTM	0.01934	1.7668	1.3849	0.9818
WOA_LSTM	0.01468	1.3886	1.0810	0.9888
SCA_LSTM	0.01809	1.5513	1.3042	0.9860
MetaREC_LSTM	0.01389	1.1797	0.9840	0.9919

**Table 7 sensors-22-07900-t007:** Results of forecasting criteria evaluated by different methods.

Method	MAPE	RMSE	MAE	$R^{2}$
BP	0.0330	3.2079	2.4763	0.9367
LSTM	0.0313	3.0010	2.4797	0.9446
WOA_LSTM	0.0162	1.6546	1.2513	0.9836
SCA_LSTM	0.0159	1.6620	1.1427	0.9831
MetaREC_LSTM	0.0151	1.6087	1.1011	0.9840

## Data Availability

Not applicable.

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
