# Peer review of "Individualized Short-Term Electric Load Forecasting Using Data-Driven Meta-Heuristic Method Based on LSTM Network"

_sensors, 2022, doi:10.3390/s22207900_

Round 1

Reviewer 1 Report

The paper proposes a new short-term load forecasting methodology based Long Short-term Memory networks (LSTM). To improve the LSTM prediction accuracy, four important parameters of the recurrent neural network are identified using the improved sine cosine optimization algorithm. The topic of the paper is interesting, and the paper is well organized. The paper can be considered for publication if the authors address this reviewer’s concerns, listed below.

·        Overall, the English is good, but some phrasing can be improved, for example “In previous studies, in order to describe more clearly the previous studies on electric load forecasting, these studies…”. Also, some typos must be revised, such as line 238 and 240, line 376 and line 574 (“it can be obtained that…”).

·        While describing the optimization process in Section 3, the four LSTM parameters representing the decision variables for the improved SCA should be mentioned (line 176). Therefore, the role of the optimization process in enhancing the LSTM performance will be clarified. Of course, their detailed description can be kept in Section 4.2.

·        Despite their well-known application, a reference should be provided for the evaluation indexes (MAPE, MAE etc.).

·        Given the high complexity of the proposed methodology, the computational resources used to run the model should be specified.

·        The forecasted days chosen for testing are 19th and 20th of May 2021, which are both working days. For better emphasize the performance of the proposed framework, a weekend day or holiday should be considered as well, as it will prove the model’s versatility for various consumption behaviours. In terms of load profiling, there are various factors that influence the energy consumption and that should be taken into account for improving the forecasting accuracy, including temperature, day of the week, holidays etc. To improve their model, I recommend the authors to consider these factors for future research.

Author Response

Dear Reviewer, 
The authors would like to thank you very much for the time and efforts you spent for reviewing the manuscript. The comments and suggestions have helped us to improve the quality of our paper. Based on the reviewers' comments, the manuscript has undergone its first revision. In order to make the manuscript more logical and professional, we have carefully made the recommended changes. Obvious syntactic and grammatical errors have been corrected. We send descriptions of how we responded and included comments on the current article version (Review 1.doc). 

We hope that the above responses address your comments and answer your questions to your satisfaction. Thank you very much for your review, we really appreciate your comments.

Reviewer 2 Report

Individualized Short-Term Electric Load Forecasting Using Data-Driven Meta-heuristic Method Based on LSTM Network

The paper utilizing Long Short-Term Memory networks and an enhanced sine-cosine algorithm based on dispatching orders, this study suggests a method for anticipating short-term electric demand. These networks have the ability to store and transmit both long-term and short-term memories. Thus, this kind of investigation is in the interest of the wider community. However, I am lacking the following:

  • The abstract and keyword should not have any acronyms. The acronyms start in the introduction section.
  • More modern literature has to be indexed in the introduction section. Read in the flowing article: (Hassan, (2020). Energy, 45(58), 33111-33127.); (Ceran, et al. (2021). Applied Energy, 297, 117161.); (Hassan, et al. (2022). Energy Reports, 8, 680-695.)
  • The novelty of the paper and the problem statement have to indexed at the end of introduction section
  • What is the resolution of the used data? and what is the simulations time step resolution?

·         What is the resolution of figures 11 and 13?

·         Figure 13 and 15 shows the electric load measurements and predictions made by various models; can authors be specifying for which days during the year? Can authors provide similar figures for each season, I mean more three figures?

  • Can authors have presented monthly results? The investigated data has to be presented daily, monthly and yearly to satisfy the study target.

The manuscript can be updated with the mentioned comments.

Author Response

Dear Reviewer, 
The authors would like to thank you very much for the time and efforts you spent for reviewing the manuscript. The comments and suggestions have helped us to improve the quality of our paper. Based on the reviewers' comments, the manuscript has undergone its first revision. In order to make the manuscript more logical and professional, we have carefully made the recommended changes. Obvious syntactic and grammatical errors have been corrected. We send descriptions of how we responded and included comments on the current article version (Review 2.doc). 

We hope that the above responses address your comments and answer your questions to your satisfaction. Thank you very much for your review, we really appreciate your comments.

Reviewer 3 Report

Individualized Short-Term Electric Load Forecasting Using Data-Driven Meta-heuristic Method Based on LSTM Network

Comments:

Justify Why Sine Cosine Algorithm is used

No need to used “We”

Reduce the number of equations

Basic LSTM Process is very common. Please reduce it

What do you mean by Experimental Evaluation? No experimental work has been done. You can used Simulation results or numerical analysis

Remove part 5.1 Evaluation Setup. It is very common for all

Please do ANOVA test

Author Response

Dear Reviewer, 
The authors would like to thank you very much for the time and efforts you spent for reviewing the manuscript. The comments and suggestions have helped us to improve the quality of our paper. Based on the reviewers' comments, the manuscript has undergone its first revision. In order to make the manuscript more logical and professional, we have carefully made the recommended changes. Obvious syntactic and grammatical errors have been corrected. We send descriptions of how we responded and included comments on the current article version (Review 3.doc). 

We hope that the above responses address your comments and answer your questions to your satisfaction. Thank you very much for your review, we really appreciate your comments.

Round 2

Reviewer 3 Report

The paper can be accpted